 

# The microbiota of moon snail egg collars is shaped by host-specific factors

Karla Piedl,[1] Frank O. Aylward,[2] Emily Mevers[1]

**ABSTRACT** Moon snails (Family: Naticidae) lay eggs using a mixture of mucus and sediment to form an egg mass commonly referred to as an egg collar. These egg collars do not appear to experience micro-biofouling or predation, and this observation led us to hypothesize that the egg collars possess a chemically rich microbiota that protect the egg collars from pathogens. Herein, we sought to gain an understanding of the bacterial composition of egg collars laid by a single species of moon snails, *Neverita delessertiana*, by amplifying and sequencing the 16S rRNA gene from the egg collar and sediment samples collected at four distinct geographical regions in southwest Florida. Relative abundance and non-metric multidimensional scaling plots revealed distinct differences in the bacterial composition between the egg collar and sediment samples. In addition, the egg collars had a lower α-diversity than the sediment, with specific genera being significantly enriched in the egg collars. Analysis of microorganisms consistent across two seasons suggests that *Flavobacteriaceae* make up a large portion of the core microbiota (36%–58% of 16S sequences). We also investigated the natural product potential of the egg collar microbiota by sequencing a core biosynthetic gene, the adenylation domains (ADs), within the gene clusters of non-ribosomal peptide synthetase (NRPS). AD sequences matched multiple modules within known NRPS gene clusters, suggesting that these compounds might be produced within the egg collar system. This study lays the foundation for future studies into the ecological role of the moon snail egg collar microbiota.

**IMPORTANCE** Animals commonly partner with microorganisms to accomplish essential tasks, including chemically defending the animal host from predation and/or infections. Understanding animal–microbe partnerships and the molecules used by the microbe to defend the animals from pathogens or predation has the potential to lead to new pharmaceutical agents. However, very few of these systems have been investigated. A particularly interesting system is nutrient-rich marine egg collars, which often lack visible protections, and are hypothesized to harbor beneficial microbes that protect the eggs. In this study, we gained an understanding of the bacterial strains that form the core microbiota of moon snail egg collars and gained a preliminary understanding of their natural product potential. This work lays the foundation for future work to understand the ecological role of the core microbiota and to study the molecules involved in chemically defending the moon snail eggs.

**KEYWORDS** marine egg mass, host-symbiont relationship, core microbiota, NRPS sequencing

Animals evolved in a world replete with microorganisms, and as a result, they often partner with microorganisms to accomplish essential functions (1). These beneficial relationships are broadly defined as symbioses, and there are examples of animal–microbe symbiotic relationships in both marine and terrestrial ecosystems.

Address correspondence to Emily Mevers, emevers@vt.edu.

The authors declare no conflict of interest.

See the funding table on p. 15.

Classic examples include the partnership between the Hawaiian bobtail squid (*Euprymna scolopes*) and its bioluminescent bacterial symbiont, *Aliivibrio fischeri*; pea aphids (*Acyrthosiphon pisum*) and their obligate symbiont *Buchnera aphidicola*; and the bryozoan *Bugula neritina* and its obligate symbiont *Candidatus* Endobugula sertula (2–4). In addition, recent reviews have comprehensively summarized many of these systems (2, 5, 6). Symbiotic relationships can broadly be categorized as mutualistic (jointly beneficial), commensalistic (beneficial to one without affecting the other), or parasitic (beneficial to one while harming the other), and the microorganisms can be either facultatively or obligately symbiotic (7). Over time, facultative symbionts can shift to obligate symbionts; this typically coincides with a reduction in the symbiont's genome size due to the nutrient-rich environment of the host, the bacterium's relatively small population sizes, asexual reproduction, and innate deletional bias (8, 9).

Symbiotic bacteria provide a variety of advantages to the host organism, including promoting host immunity, providing essential nutrients, and producing chemical defenses (10–12). For example, the marine tube worms (Siboglinidae) are gutless and rely on endosymbiotic thiotrophs and methanotrophs for primary metabolism, leafhoppers (Cicadellidae) rely on its two symbionts, *Ca. Paulibaumannia cicadellinicola* and *Ca. Karelsulcia muelleri* for biosynthesis of essential vitamins and amino acids, respectively, while solitary wasps in the Philanthini tribe (Crabronidae) apply *Ca. Streptomyces philanthi* to the underground brood chamber to chemically protect their larvae from pathogenic fungi (12–14). In addition, analysis of symbionts with reduced genomes can provide insights into the ecological role they provide the host. Genomic analysis of *Burkholderia gladioli* Lv-StB, an obligate symbiont associated with the African beetle *Lagria villosa*, revealed that it lacks the genes necessary to produce many amino acids, yet maintains the genes necessary to produce lagriamide, an antifungal polyketide. In fact, the biosynthetic gene cluster (BGC) responsible for producing lagriamide consumes 4.2% of the entire genome (15). It has been hypothesized that *B. gladioli* Lv-StB produces lagriamide to protect the eggs from fungal infections. The biosynthetic potential for defensive natural products within the host–microbe systems are known to represent up to 20% of the bacterial symbiont's entire genome, indicating the critical ecological role of these molecules (16).

Marine egg masses, particularly those laid by invertebrates, are highly nutrient rich and lack obvious protection mechanisms, such as parental protection. Despite this, field observations suggest they are not overrun with microbial pathogens (bacteria or fungi) and are not commonly preyed upon. It has previously been suggested that egg masses harbor symbiotic bacteria to chemically protect them from predation and/or infection (17, 18), which is supported by previous reports indicating that treatment of egg masses (e.g., *Palaemon macrodactylus* and *E. scolopes*) with antibiotics sensitizes them to infections by both bacterial and fungal pathogens (18–21). However, very few studies investigated the microbiota components of these systems or their defensive agents (19, 20, 22). Most recently, it was discovered that the Hawaiian bobtail squid and the nudibranch *Dendrodoris nigra* contain specialized glands that harbor bacteria-rich jelly that is mixed with eggs to form egg masses, known as clutches (23–25). The Hawaiian bobtail squid's mucus is known to be enriched in Alphaproteobacteria, and chemical investigations into bacterial strains isolated from the egg clutches revealed the production of antimicrobial natural products. It has been proposed that these natural products serve a defensive role and protect the egg clutches from fungal infections by *Fusarium keratoplasticum*, a specialist pathogen (21).

Moon snails (family: Naticidae) are a large family of predatory snails with estimates of over 270 distinct species distributed worldwide (26). Limited studies into moon snail biology have focused primarily on species identification, predation habits, and egg production, but much of what is known about their life cycle was gleaned from studying a few species (26–33). Similar to the Hawaiian bobtail squid, moon snails coat their eggs with a gelatinous mucus; however, they combine this mixture with sediment to form egg masses that are commonly referred to as a "sand" or "egg" collars (26). The individual

eggs are contained in capsules surrounded by a mucus distinct from the one used to bind the egg collars. It is currently unknown if either mucus contains bacteria like the other Mollusca systems. Each egg collar has tens to thousands of individual egg capsules arranged in rows or in a zig-zag feature (33). Some moon snails lay thicker egg collars, and their egg capsules are visible to the naked eye, while other species lay thinner egg collars where the individual egg capsules are not visible, even when examined under light. The details on embryogenesis are even less clear for most species of moon snails. It has been proposed that eggs are laid around full moons and take between 2 to 10 weeks to develop into larvae (27, 28). As the egg collars age, they lose their structural integrity, and this is thought to occur as the larvae are released and is a result of the mucus layer degrading (29–32). Most moon snail species are believed to hatch into free-swimming veliger larvae, but some hatch pediveliger juveniles, though the pediveliger juveniles are found mainly in higher latitudes following Thornson's Rule (28).

Field observations suggest that the egg collars do not experience micro-biofouling or extensive predation, and therefore, we have hypothesized that the egg collars harbor chemically rich bacterial symbionts. Previous work by our group has taken a culture-based approach to understand the chemical potential of culturable bacteria associated with egg collars from moon snails (*Neverita delessertiana*) collected in SW Florida, leading to the generation of a library with >700 bacterial strains. Chemical and biological investigations into these strains led to the identification of a specific natural product, pseudochelin A, produced by numerous bacterial strains associated with the egg collars collected in SW Florida, which inhibits biofilm formation by both ecologically relevant and pathogenic Gram-positive bacteria (34). However, it is unlikely that our bacterial strain library resembles the core microbiota of the moon snail egg collars as the vast majority of the bacterial strains, especially true bacterial symbionts, are commonly unculturable under typical conditions (35). Herein, we took a cultivation-independent approach to both understand the bacterial composition of the moon snail egg collars as it relates to the sediment, geographical location, and seasonality and to gain insights into its true biosynthetic potential.

## RESULTS

### Bacterial community composition of moon snail egg collars and corresponding sediment

To begin to understand the microbiota composition of the moon snail (*N. delessertiana*) egg collars, egg collar and sediment samples were collected from four distinct environmental niches across Matlacha Pass, Pine Island, FL, in January 2023 (Fig. S1). The sites included a sandbar open to the sound (site 1), two shallow mangrove bays with muddy bottoms on either side of the sound (sites 2 and 3), and a sandy-bottom beach off the northern point of Pine Island (site 4). Sediment samples were taken directly under each collected egg collar, as it is believed that the moon snail uses this during the egg-laying process. Sequencing of the 16S rRNA gene from the egg collar and sediment DNA extractions resulted in 12,784,880 total reads across 16,384 unique sequences. Clustering against the Silva reference database (v. 138.1) grouped the amplicon sequence variants (ASVs) into operational taxonomic units (OTUs). Plotting the relative abundances of the top 20 taxa showed a clear delineation between the egg collar and sediment samples for three of the four collection sites (Fig. 1A; Fig. S2). The only exception was site 1, where the sediment and egg collar samples have visually similar taxonomic compositions.

### Clustering analysis to examine sample relatedness

Relatedness between sites and sample types was analyzed by non-metric multidimensional scaling (NMDS) using Bray–Curtis distances to create distance matrices (Fig. 1B; Fig. S3). Analysis of egg collar and sediment samples from all sites revealed that sample type is the main variable driving sample clustering at all levels of phylogeny, meaning egg collars are more similar to other egg collars than to sediment samples. The clustering of

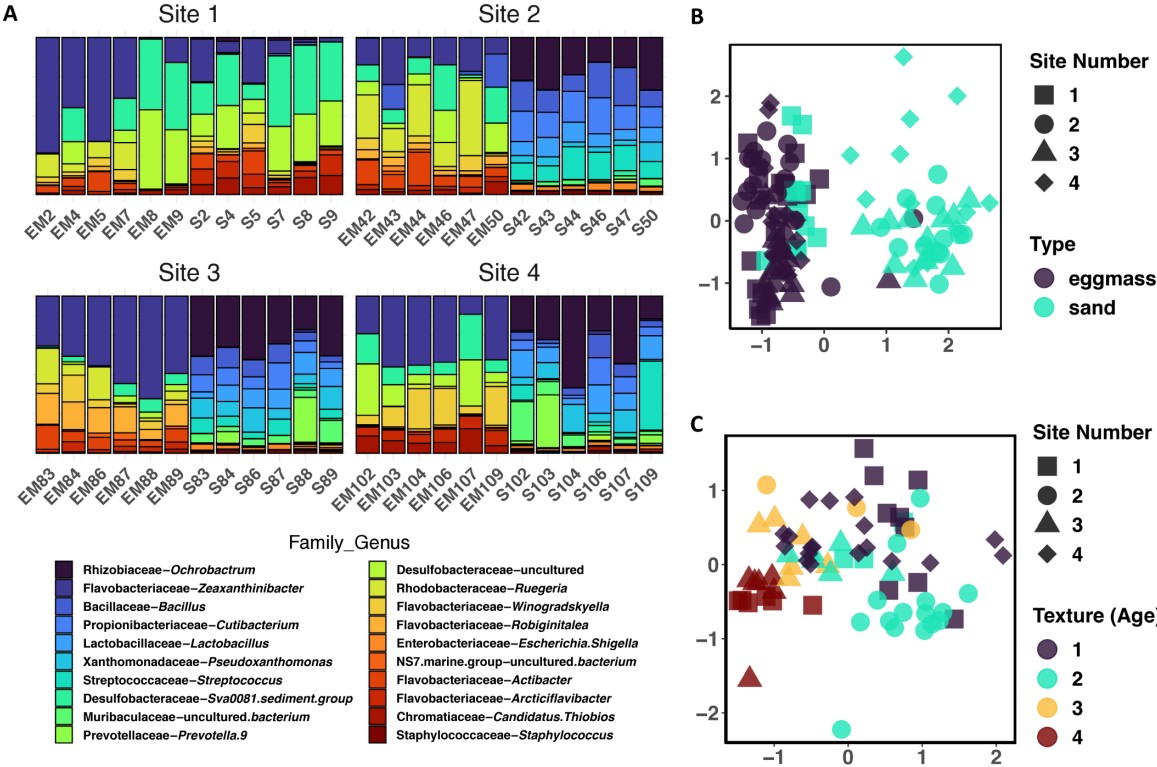

**FIG 1** (A) Relative abundance of bacterial genera from the egg collar and sediment samples. Each egg collar was sequenced in triplicate, and sediment was sequenced in duplicate. The relative abundance was calculated from the average of the sequencing runs. Different colors indicate different genera, and height of bars indicate relative abundance of genera in that sample. (B) Non-metric multidimensional scaling (NMDS) plot of Bray–Curtis diversity distances of egg collar and sediment samples as a function of site location. Color indicates sample type, and site is indicated by shape. (C) NMDS plot of Bray–Curtis diversity distances for texture of egg collar samples as a function of site location. Different textures are indicated by color, firm (dark purple), less firm (light green), crumbly (yellow) and fragile (red), and site number is indicated by the same shapes as in (A).

the egg collar samples was statistically significant at all levels below phylum ($P < 0.001$) and was strongly significant at the phylum level ($P = 0.005$) (Fig. S3; Table S1). However, there was more variability within the sequences from the sediment samples. Only the sediment samples from site 1 consistently clustered with the egg collar samples from all sites; sediment samples clustered by site at both the phylum and genus levels (permutational multivariate analysis of variance [PERMANOVA], $P < 0.001$) (Table S1; Fig. S4). Similarly, NMDS analysis of just egg collar samples across all sites confirmed that these samples are statistically similar at both the phylum and class levels ($P = 0.079$ and $P = 0.019$, respectively). In addition, the egg collar samples did not cluster by site, color, or size of the egg collar. During field collections, each egg collar was classified as either fragile (i.e., breaking apart when picked up) or firm (i.e., did not rip easily). As previously reported, the texture of the egg collars changes over time (28, 33); therefore, we believe that the firm egg collars represent those that have recently been laid, and the egg collars become more fragile as the embryos develop (Table S2) (33). Interestingly, a significant difference was noted between the microbial communities of egg collars with different textures (PERMANOVA, $P < 0.001$) (Fig. 1C; Table S1).

## Changes in microbiota composition across time

Additional egg collars were collected the following year from site 2 across 3 months, Dec. 2023, Jan. 2024, and twice in Feb. 2024. By chance, one of the Feb. 2024 collections occurred 2 days after a full moon, with a noticeable increase in the number of firm egg collars; thus, field observations lead to the hypothesis that moon snail egg collars may be laid around full moons. Ten egg collars were collected each month, while 10 firm and

fragile egg collars were collected on the second day in Feb. 2024. All collected egg collars were processed using the same platform described above and yielded 7,885,634 reads across 9,539 unique sequences. The relative abundance of the top 20 genera was plotted for egg collars collected in the 2023–2024 season (Fig. S5A), and this showed that the composition of the microbiota is consistent across all 3 months. In addition, the overall composition of the sequences was similar between the 2 years. The only noticeable difference between the months was the presence of a higher percent abundance of *Pseudoalteromonas* sp. in January compared with February (fourfold, $P$ = 0.0018). An NMDS plot of the targeted collection of fragile and firm egg collars using Bray–Curtis dissimilarity distances showed a clear difference in genera abundance between the two sample types (Fig. 2A; Table S3), with the most striking difference being a higher total abundance of the Desulfobacteraceae "Sva0081 sediment group" (Sva0081) on firm egg collars compared with fragile egg collars (112-fold, $P$ = 1.6e$^{-11}$). While fragile egg collars had a higher abundance of two Alphaproteobacteria genera - *Ruegeria* (eightfold, $P$ = 3.9e$^{-11}$) and *Loktanella* (sixfold, $P$ = 1.1e$^{-10}$). NMDS plot comparing the relationship

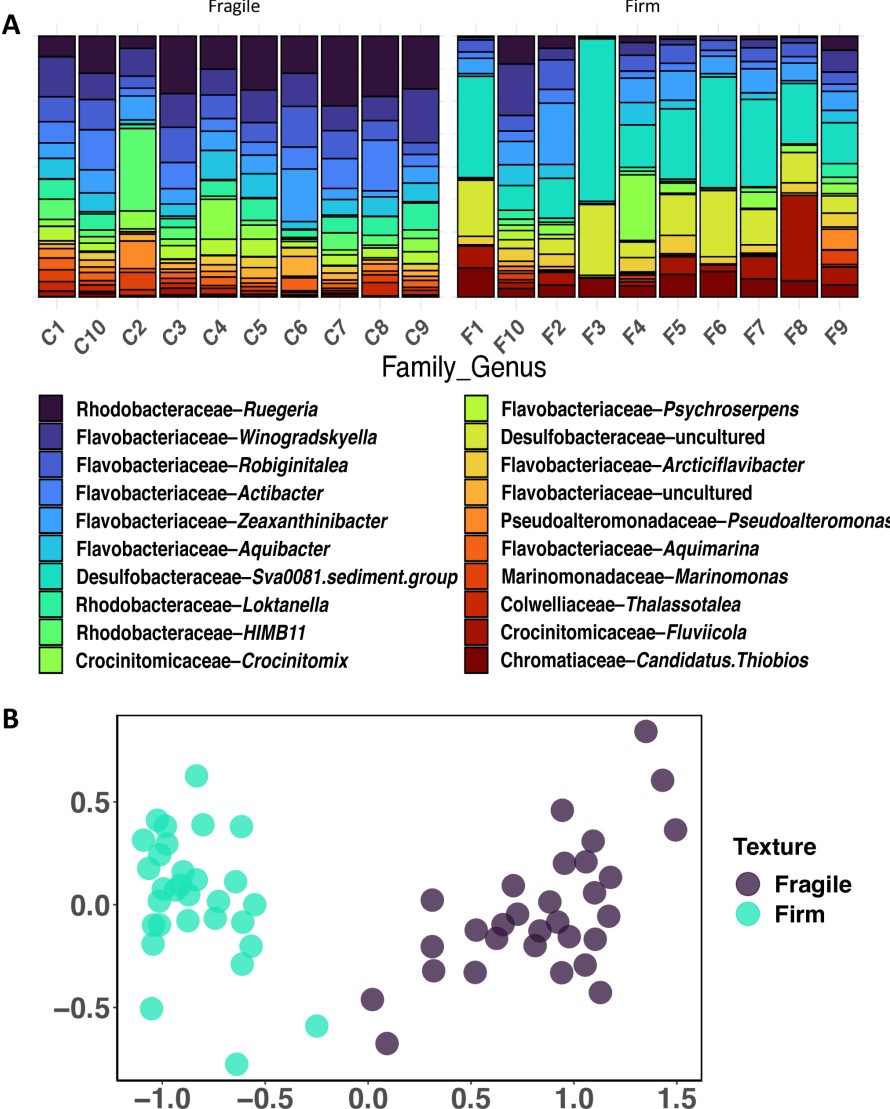

**FIG 2**  (A) Relative abundance of top 20 genera from fragile (C) and firm (F) egg collars, calculated as an average of the sequencing rungs. (B) NMDS plot of Bray–Curtis diversity distances for fragile and firm collar samples collected in late February.

between these firm and fragile egg collars formed two clusters corresponding to the two texture types (PERMANOVA, $P < 0.001$) (Fig. 2B).

## Core microbiota of the moon snail egg collars

The counts for the 50 most abundant genera present in the egg collar and sediment samples were log-transformed and plotted as a heatmap with hierarchical clustering, which clearly shows differences in genera abundance between sample types (Fig. 3). Sediment samples from sites 2–4 formed a distinct branch in the dendrogram. This is likely driven by the fact that many of the genera that are highly enriched in these sediment samples are either absent or present in low abundance in the egg collars, including *Ochrobactrum* (Rhizobiaceae), *Bacillus* (Bacillaceae), and *Cutibacterium* (Propionibacteriaceae). Sediment samples from site 1, which more closely resemble egg collar samples, clustered with egg collar samples in the dendrogram as these sediment samples contain lower read counts of *Ochrobactrum* and *Cutibacterium* and higher

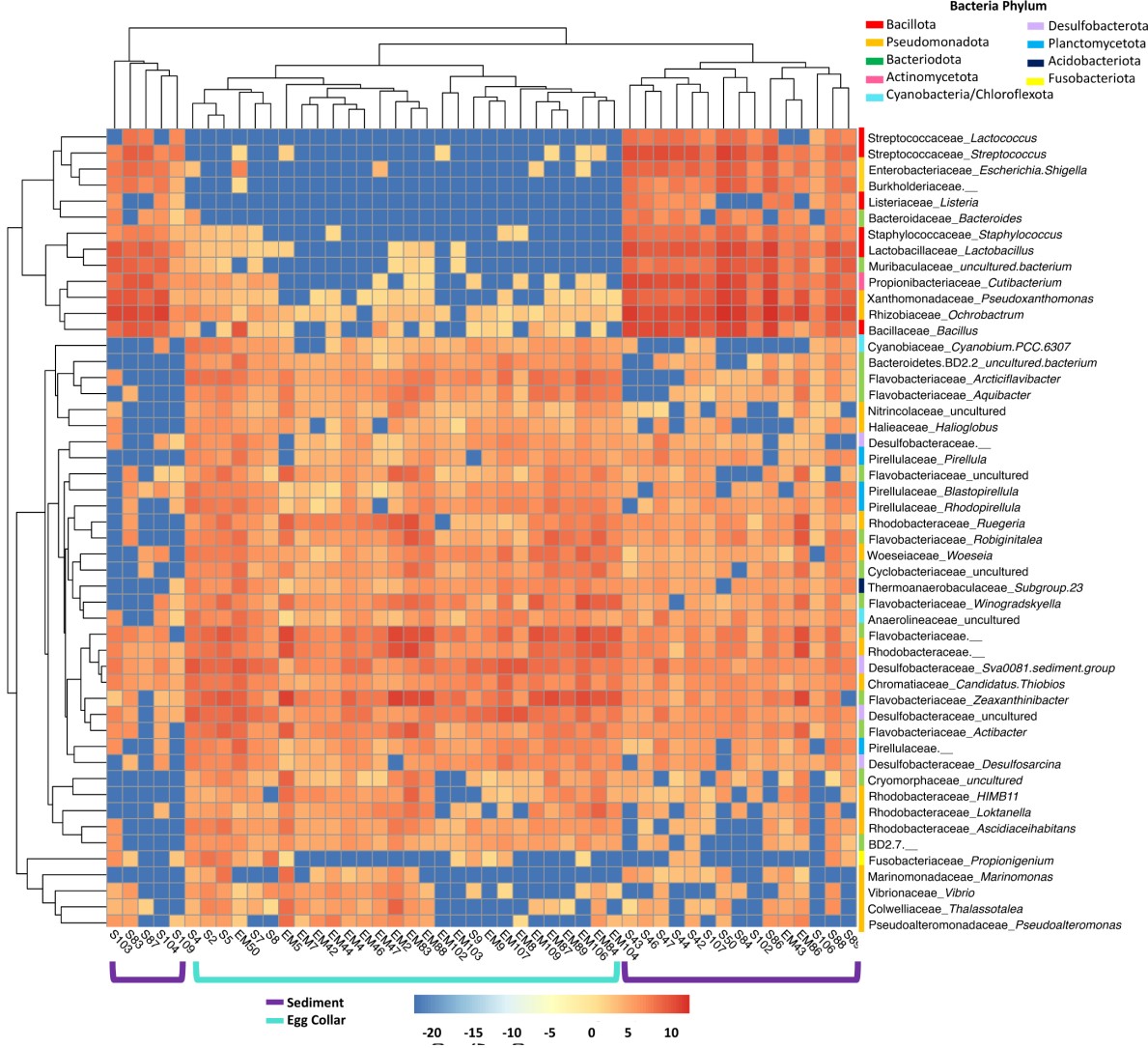

**FIG 3** Heatmap of abundance counts from top 50 genera after log transformation using pheatmap in RStudio. Euclidean distances were used for hierarchical clustering of samples and genera. Sample number and type are shown on the y-axis, aqua brackets indicate egg collar samples and sediment samples from site 1, dark purple brackets indicate samples from sites 2–4. Family_Genus is shown on the y-axis, a colored bar to the left of the taxa indicates the phyla for each OTU: Bacillota, Pseudomonadota, Bacteriodota, Actinomycetota, Cyanobacteria/Chloroflexota, Desulfobacterota, Planctomycetota, Acidobacteriota, and Fusobacteriota are indicated by red, gold, green, pink, light blue, light purple, blue, navy, and yellow bars, respectively.

counts of *Propionigenium* (Fusobacteriaceae). Egg collars formed a second branch in the dendrogram, with sediment samples from site 1 forming a sub-branch. There are clear differences between the egg collar branch and the two sediment sub-branches, including the enrichment of many Flavobacteria genera, such as *Winogradskyella*, *Zeaxanthinibacter*, and *Aquibacter*. This, along with the relative abundance and NMDS plots (Fig. 1A through C), strongly suggests that the moon snail egg collars possess a core microbiota that is distinct from the sediment in which they are laid.

Genera abundance was used to assess and identify the core microbiota of the egg collars. To begin to define the composition of the core microbiota of the egg collars, we analyzed the ASVs present on at least 80% of the egg collar samples (core80) (e.g., 19 of 24 total egg collar samples), resulting in 22 OTUs (Table 1). Eight of the identified genera were found in high abundance (>50,000 total reads) and made up 7.2% of total reads: the Flavobacteriaceae *Actibacter, Robiginitalea, Winogradskyella,* and *Zeaxanthinibacter*, an uncultured bacterium from the NS7 Marine group*, the Gammaproteobacteria *Ruegeria,* Sva0081 and an uncultured member from the Desulfobacteraceae (Table S4). The variable core of the egg collars (present on 50%–80% of egg collars) collected in 2023 is mostly composed of members from the Pseudomonadota: 17% is composed of Alphaproteobacteria and 24% is made up of Gammaproteobacteria (Table S5). When compared with the microbiota of sediment samples, it is clear that the egg collar microbiota is distinct because only the NS7 marine group (Flavobacteriia) is present in both (Table 1; Table S6). To analyze annual differences, samples collected in Dec. 2023 and Jan. and Feb. 2024 were used to determine the core80 (and variable core) microbiota for 2024 collection (Table S7). Flavobacteriia remained largely consistent between years, making up 58% of the 2024 core80 (compared with 36% of the 2023 core), and *Ruegeria* (Alphaproteobacteria) is also present in both years. The core80 of firm versus fragile egg collars was analyzed, resulting in 18 and 13 OTUs for firm and fragile egg collars, respectively. The core80 from egg collars of both textures were dominated by Flavobacteriia, comprising 42% of the firm egg collar core80 and 80% of the fragile egg collar core80 (Tables S8 and S9). Firm egg collars were also dominated by Sva0081, comprising up to 20% of the total abundance. As the texture changes, Sva0081 total abundance decreases to less than 1% ($P = 2.69e^{-05}$), while the average Flavobacteriia total abundance increases from 20% to 31% ($P = 0.0148$).

## Comparison of taxonomic richness between egg collar and sediment samples

Next, analysis of species richness (after rarefaction at 500 sequences) was used to explore the α-diversity within each sample type (Fig. 4A; Fig. S6). Species richness and Hill values show that egg collar samples had overall lower species richness and contained fewer taxa than the sediment samples (Table S10), suggesting host filtering for a specific subset of total microbial diversity due to the environment of the egg collars. Surprisingly, sediment samples from site 1 had on average the highest species richness values, indicating these samples are enriched in low occurrence genera even though they clustered with the egg collar samples. Further analysis of the taxonomic enrichment within each sample type using DESeq2 on ASV counts identified ASVs that were significantly enriched in each sample type (Fig. S7). Applying a strict *P*-value ($<10e^{-6}$), five genera that all belonged to the Flavobacteriia (*Winogradskyella, Aureitalea, Aquibacter, Muriicola*, and *Zeaxanthinibacter*) were found to be enriched on the egg collar samples (Fig. 4B). Conversely, sediment samples were found to be enriched in 2.5× more bacterial genera, including *Ochrobactrum, Cutibacterium, Pseudoxanthamonas, Streptococcus*, and *Bacteroides* (Fig. 4C).

## Sequence diversity of the adenylation domain (AD) within moon snail egg collars

Nonribosomal peptide synthetases (NRPSs) are large multienzyme machineries that assemble diverse nonribosomal peptides (NRPs), a subset of natural products that commonly incorporate a significant amount of post-modification. The adenylation

**TABLE 1** Lineage of genera found in core80 microbiota of the moon snail egg collars

| Phylum[a] | Class | Order | Family | Genus | Count |
|---|---|---|---|---|---|
| Acidobacteriota (Acidobacteria) | Thermoanaero-baculia | Thermoanaero- baculales | Thermoanaero- baculaceae | Subgroup 23 | 21 |
| Bacteroidota (Bacteriodetes) | Bacteroidia | Bacteroidales | Bacteroidetes BD2-2 | Uncultured[b] | 21 |
| | Cytophagia | Cytophagales | Cyclobacteriaceae | Uncultured[b] | 24 |
| | Flavobacteriia | Flavobacteriales | Flavobacteriaceae | *Actibacter* | 24 |
| | | | | *Aquibacter* | 22 |
| | | | | *Arcticflavibacter* | 24 |
| | | | | *Robiginitalea* | 23 |
| | | | | *Winogradskyella* | 22 |
| | | | | *Zeaxanthinibacter* | 24 |
| | | | | Uncultured[b] | 21 |
| | | | NS7 Marine group | Uncultured[b] | 21 |
| Chloroflexota (Chloroflexi) | Anaerolineae | Anaerolineales | Anaerolineaceae | Uncultured[b] | 20 |
| Desulfobacterota[d] | Desulfobacteria[a] | Desulfobacterales | Desulfobacteraceae | Sva0081 | 24 |
| | | | | Uncultured[b] | 24 |
| Planctomycetota | Planctomycetacia | Pirellulales | Pirellulaceae | *Pirellula* | 20 |
| | | | | *Rhodopirellula* | 19 |
| Pseudomonadota (Proteobacteria) | Alphaproteo- bacteria | Rhodobacterales | Rhodobacteraceae | *Ascidiaceihabi- tans* | 23 |
| | | | | HIMB11 | 20 |
| | | | | *Ruegeria* | 20 |
| | Gammaproteo- bacteria | Chromatiales | Chromatiaceae | *Ca*. Thiobus[c] | 23 |
| | | Unknown[e] | Unknown | Uncultured[b] | 19 |
| | | Steroidobacterales | Woeseiaceae | *Woeseia* | 21 |

[a]Taxonomy reflects the current accepted bacterial nomenclature; previously used names are indicated in parenthesis (36–38).
[b]Uncultured indicates that the organism has not been cultured in a laboratory setting, and that taxonomy has not been assigned (as of Silva v. 138–2019).
[c]Candidatus indicates the genus has been described, but no species has been cultured in the laboratory.
[d]Phylum still has candidatus status (39).
[e]Originally labeled *incertae sedis* (Latin for 'of uncertain placement').

domain (AD) is a component of the NRPS gene and is responsible for selecting and loading amino acids. AD sequences are highly conserved and have previously been used to provide insights into the biosynthetic potential of a microbiota (40). Using general primers specific for ADs, we amplified and sequenced the ADs from the Feb. 2023 egg collar DNA samples, generating 854,930 raw reads. Raw sequences were filtered through a similar platform as the 16S sequences, reducing the total number to 19,085 counts of 2,720 non-redundant AD sequences.

To gain an understanding of the potential BGCs that these AD sequences represent, we analyzed the AD sequences against our internal library of genomes, the Minimum Information about a Biosynthetic Gene Cluster (MIBiG) database, and the NCBI non-redundant protein database. First, alignment of the AD sequences with our internal library, small database of sequenced genomes (36) representing bacterial strains isolated from the moon snail egg collars, failed to produce any high confidence matches. Next, we took AD sequences with a minimum of 20 counts and aligned them with the MIBiG database using blastx (50% sequence identity cutoff, e-value $<1\times10^{-10}$). Analysis of the 50 most abundant sequences (representing 76% of the total count) with the MIBiG database revealed that 70% are not associated with known NRPS BGCs. Of the 13 sequences that matched to ADs within MIBiG BGCs, all matched to either bacitracin, cephamycin, or thaxteramide (Table S11). Finally, we aligned the AD sequences against the NCBI non-redundant protein database with the blastx alignment function, resulting in 8,684 unique matches. Filtering the matches with a strict e-value ($<10\ e^{-38}$) resulted in 23 distinct matches to annotated NRPS genes (Table 2). Full genomes for these matches were downloaded, if available (Table 2), and analyzed by AntiSMASH to identify the specific BGC to which the gene was mapped (41). Of the 23 matches, 12 are associated

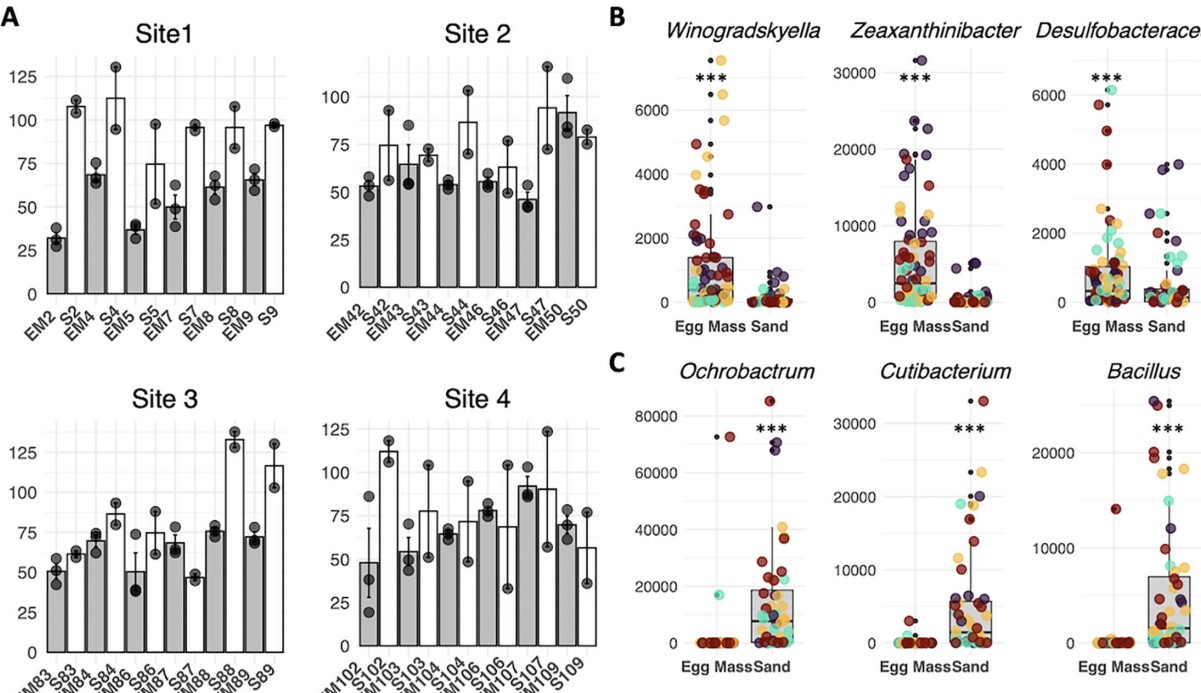

**FIG 4** (A) Alpha diversity as richness after rarefaction in egg collars and sediment samples collected from the four different sites. Bar plots showing averages of collected samples. Each egg collar sample (light gray) is shown directly to the left of its corresponding sediment sample (white). Error bars indicate the standard error, and each datapoint is shown in gray. (B) Box plots of genera sequence counts of *Winogradskyella* ($P = 1.01e^{-08}$), *Zeaxanthinibacter* ($P = 3.57e^{-12}$), an uncultured Desulfobacteraceae ($P = 6.16e^{-03}$) in egg collars. (C) Box plots of genera sequence counts of *Ochrobactrum* ($P < 1e^{-05}$), *Cutibacterium* ($P < 1e^{-05}$), and *Bacillus* ($P = 5.99e^{-14}$), with high abundance in sediment. *P*-values calculated with the Wilcoxon test for non-normally distributed data as identified with the Sharpiro–Wilk normality test. Different colored dots represent the four sites (yellow = site1, orange = site 2, purple = site 3, and black = site 4).

with BGCs with similarity to known natural products BGCs. Three identified natural product BGCs (corbomycin, nostopeptolide, and microsclerodermin) each matched two distinct sequences, one sequence matched with 73% identity to coprisamide BGC. Each of these natural product BGCs contain between 5 and 10 ADs; therefore, to investigate these matches even further, we attempted to map additional AD sequences (regardless of e-value) to the ADs of these three pathways. ADs for all four BGCs were identified using antiSMASH, then the sequences were aligned to the egg collar AD sequences using ClustalW. This led to the identification of additional sequences matching all ADs within these BGCs with percent alignment scores of 47.4%–72.4% for nostopeptolide, 49.6%–81.9% for microsclerodermin, 51.6%–78.3% for corbomycin, and 44.1%–83.1% for coprisamide (42), suggesting that these or analogous compounds may be ecologically relevant (Fig. 5).

## DISCUSSION

The results outlined above support the hypothesis that *N. delessertiana* moon snail egg collars harbor a core bacterial microbiota that is distinct from the sediment in which they are composed. Using 16S rRNA sequencing, we showed that the bacterial composition of the egg collars is both distinct and less rich than the bacteria found within the sediment. The observed difference in α-diversity between the egg collars and sediment samples is indicative of species filtering by the host and is expected to occur during the process of establishing a symbiotic relationship (43). This phenomenon has been observed in a wide range of symbiotic associations, such as when sorghum (*Sorghum bicolor*) establishes symbiosis with root-colonizing arbuscular mycorrhizal fungi or in the establishment of the leaf microbiome of seagrass (*Zostera marina*) (44). The moon snail incorporates microbial-rich sediment into its egg collars (26, 45), and so it is notable that

**TABLE 2** Sequenced adenylation domain alignments with NCBI non-redundant protein database (including gene alignment e-value, accession number and organism) and AntiSMASH biosynthetic gene cluster identification and metabolite prediction

| E-value[a] | Gene accession | Source organism | Genome Accession[b] | Metabolites[c] |
|---|---|---|---|---|
| 3.28e-39 | WP_280671808 | *Kitasatospora* sp. MAP12-44 | GCA_029892095 | Coprisamide (73%) |
| 6.51e-39 | WP_184933496 | *Kitasatospora kifunensis* | GCA_014203855 | Corbomycin (25%) |
| 6.95e-39 | WP_054047858 | *Alloactinosynnema* sp. L-07 | GCA_900070365 | Hexacosalactone A (9%) |
| 8.01e-39 | WP_184933498 | *Kitasatospora kifunensis* | GCA_014203855 | Corbomycin (25%) |
| 8.60e-39 | WP_038841356 | *Salinispora arenicola*[d] | | |
| 8.77e-39 | WP_019901716 | *Salinispora arenicola*[d] | | |
| 8.77e-39 | WP_028182853 | *Salinispora arenicola*[d] | | |
| 9.73e-39 | WP_184944425 | *Kitasatospora kifunensis* | GCA_014203855 | Feglymycin (31%) |
| 1.03e-45 | WP_009809411 | *Roseobacter* sp. MED193 | GCA_000152965 | NA[e] |
| 9.31e-42 | WP_008207429 | *Roseobacter* sp. SK209-2-6 | GCA_000169455 | NA[e] |
| 5.14e-41 | EEW58188 | *Ruegeria* sp. TrichCH4B | *GCA_000161815*[f] | |
| 1.66e-40 | WP_011539722 | *Ruegeria* sp. TM1040 | GCA_000014065 | NA[e] |
| 2.65e-39 | NND55032 | Gammaproteobacteria bacterium | GCA_013001905 (MAG)[g] | NA[e] |
| 2.89e-39 | WP_190467934 | *Anabaena azotica* FACHB-119 | GCA_014697625 | Nostophycin (27%) |
| 3.09e-39 | WP_190905042 | *Anabaena catenula* FACHB-362 | GCA_014698735 | Nostopeptolid A1 (50%) |
| 3.57e-39 | WP_013940237 | *Corallococcus macrosporus* DSM 14697 | GCA_002305895 | Microsclerodermin (35%) |
| 4.51e-39 | WP_026723918 | *Fischerella* sp. PCC 9431 | *GCA_000447295*[f] | |
| 4.60e-39 | WP_204817706 | *Corallococcus macrosporus* DSM 14697 | GCA_002305895 | Microsclerodermin (35%) |
| 5.03e-39 | WP_027842330 | *Mastigocoleus testarum* BC008 | GCA_001456025 | NostopeptolideA2 (50%) |
| 5.50e-39 | WP_187223241 | *Actinokineospora xionganensis* | GCA_014323725 | Sandarazol A (12%) |
| 8.01e-39 | WP_240359913 | *Pyxidicoccus trucidator* | GCA_010894435 | NA[e] |
| 9.99e-39 | AUZ99700 | *Fischerella* sp. F29[h] | | |
| 1.36e-39 | HAG81025 | Cyanobacteria bacterium UBA12227 | GCA_003450835 (MAG)[g] | Cyanochelin A (38%) |

[a]E-value and percent identity are based on alignment of the query sequence with protein.
[b]All genomes listed in this column were downloaded unless otherwise noted.
[c]Similarity of metabolite predicted by identified BGC to known BGC; percent identity between BGCs indicated in parenthesis.
[d]Unable to determine specific strain of *Salinispora arenicola* from gene accession number.
[e]NA indicates sequence did not match ADs identified by AntiSMASH.
[f]Italics indicate the genome has been suppressed by NCBI and were not downloaded.
[g]Genomes obtained are metagenomic assembled genomes (MAGs).
[h]No genome was available for this organism.

the bacterial composition of the egg collars is distinct from this sediment. This bacterial composition is highly consistent across all collection sites, despite large variations in the microbiota within the sediment samples, particularly between the mangrove bays (sites 2 and 3) and the sandbar (site 1). In addition, our analysis indicates that the egg collar core80 microbiota consists of Flavobacteriia (36%), Alphaproteobacteria (13.6%), and Gammaproteobacteria (13.6%), which is quite distinct from the core microbiota of the sediment samples. This suggests that the egg collars are actively filtering the bacterial composition, shaping the core microbiota through topographical features, presence of specific nutrients, or suppression by the egg collars (46, 47). Finally, it is worth highlighting that the core microbiota is taxonomically dissimilar from our curated strain library, which is dominated by Gammaproteobacteria (33%), Bacilli (32%), and Alphaproteobacteria (17%), with Flavobacteriia representing only 6% of our strain library (34). This dissimilarity between our culture library and metataxonomic results is likely a result of known biases in culture-dependent methods (48).

Furthermore, we found that the egg collar microbiota significantly correlates with egg collar texture (i.e., firm to fragile). The egg collar texture corresponds to age of the egg collars, with firmer egg collars being laid more recently compared with the fragile egg collars as the mucus degrades (28, 33). There was a clear difference in the relative abundance of genera between the firm and fragile egg collars, which was supported by clustering on NMDS plots. Interestingly, the core microbiota (core80 or variable) of the sediment samples did not resemble the core80 of firm egg collars, despite likely being freshly laid with sediment. As the egg collars aged, there was a

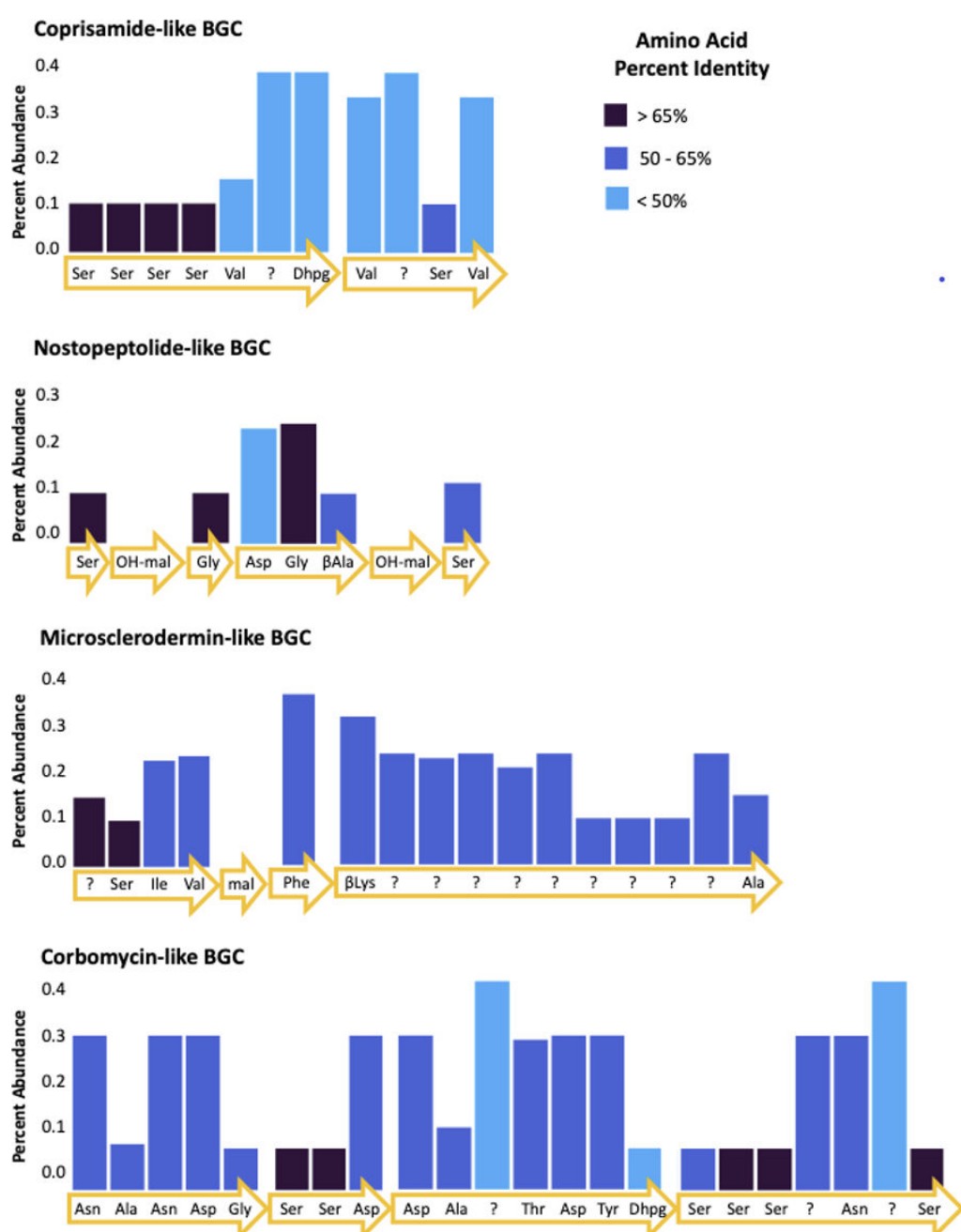

**FIG 5** Percent abundance of sequences matching adenylation domains in the identified natural product BGCs. Percent identity (amino acid) between adenylation domains and sequences indicated by color. Arrows indicate genes with predicted amino acids recognized by adenylation domains (AntiSMASH v. 7.1.0). Malonate (mal) and hydroxy-malonate (OH-mal) are PKS domains and were not part of the sequencing.

large increase in the average abundance of Flavobacteriia and a significant decrease in abundance of Sva0081, an obligate anaerobe, suggesting a loss of an oxygen gradient over time. Mucus produced by marine molluscs is a complex biopolymer thought to contain an acidic polysaccharide glycosaminoglycan component (49). Flavobacteriia are well known for their proficiency in degrading carbohydrates and large biopolymers, including chitin, pectin, lignocellulose, and cellulose (50–52). Genomes from marine Flavobacteriia are enriched in polysaccharide utilization gene clusters and enzymes that degrade complex carbohydrates and glycoconjugates (53, 54). Therefore, it is possible

that either the mucus is the source of the microbiota components, or the nutrient profile of the mucus quickly selects for specific bacterial strains. Additional studies are needed to fully understand the factors that lead to the selection of Flavobacteriia.

Comparison of the moon snail egg collar microbiota composition to other Mollusca shows that the egg collar microbiota is distinct from that of the squids, but somewhat similar to whelks. The microbiota composition within the accessory nidamental gland (ANG) of the bobtail squid is dominated by Alphaproteobacteria (66.1%) and Verrucomicrobia (29.1%), with minor contributions from Gammaproteobacteria (3.8%) and Flavobacteriia (1.4%) (55). This is highly similar to the reported microbiota composition of the opalescent in-shore squid *Doryteuthis opalescens* egg masses (56). More recently, analysis of the Kellet's whelk (*Kelletia kelletii*) perivitilline fluid, which houses embryos in a protein–polysaccharide matrix, revealed that although Alphaproteobacteria comprise the majority of the core (57%), Flavobacteriia represent a significant component (28%), which is similar to the core microbiota of the moon snails (57).

Analysis of sequence diversity of the AD within the associated bacterial microbiota of the egg collars has provided further insights into its natural product potential. ADs are core domains within NRPSs, which encode non-ribosomal peptides (NRPs). NRPs are structurally diverse peptides that can incorporate over 300 non-proteinogenic amino acids (58). Bacteria are known to produce a diverse array of biologically active NRPs, including natural products with potent antibacterial, anticancer, and immunosuppressant activities (59). The majority of the AD sequences (76%) within the egg collars failed to match with genes in the MIBiG database, suggesting that they represent new natural products or belong to natural products with unknown BGCs. However, as the MIBiG database is not comprehensive, blastx analysis performed on all egg collar AD sequences led to high confidence matches (e-value $<10e^{-38}$, percent identity >50%) to ADs within nostopeptolide, microsclerodermin, corbomycin, and coprisamide-like BGCs. This was supported by the presence of additional AD sequences matching multiple ADs within each pathway, thus suggesting that the egg collars may possess the ability to produce these, or highly related molecules. On-going efforts are focused on understanding the natural product potential of the core microbiota within the egg collars using metagenome-assembled genomes and metabolomic studies.

## Conclusion

This study indicates that the moon snail egg collars possess a distinct and less rich bacterial microbiota than the sediment in which they are laid, and has identified potential NRPS BGCs; further, we suggest that symbiotic bacteria on the egg collars are transmitted from the parent. Future work should focus on the analysis of the gelatinous matrix from moon snail parents to better understand the transmission of symbionts from parent to egg collar. Additionally, a clearer understanding of the metabolites associated with the egg collars and their ecological roles can be achieved through metabolomic investigations. This work paves the way for future studies to understand the ecological importance of metabolites produced by symbiotic bacteria on moon snail egg collars.

## MATERIALS AND METHODS

### Sample collection and DNA extraction

Both sediment samples and egg collars belonging to *N. delessertiana* moon snail were collected at four sites in the Matlacha Pass in Jan. 2023 (site 1–82°–5′ 17.09″ W 26°–42′ 12.462″ N, site 2–82°–6′ 37.54″ W 26°–40′ 22.638″ N, site 3–82°–4′ 54.1″ W 26°–42′ 1.4″ N, and site 4–82°–8′ 15.51″ W 26°–42′ 10.662″ N) and from site 2 in Dec. 2023, and Jan. and Feb. 2024 with permission from the Florida Wildlife Conservation (permit number: SAL-21–2113-SR). For the Jan. 2023 collection, 10 egg collar and sediment samples were collected per site, and information on collection sites, such as water temperature, and egg collars (size, depth, color, texture, biofilm, etc.) was recorded (Table S3). Individual egg collars were collected and divided into thirds, placed in 15 mL falcon tubes, and

preserved with ethanol (EtOH). In addition, sediment directly under each egg collar was collected in two 15 mL falcon tubes and preserved in EtOH. All samples that were preserved in EtOH were stored at 4℃ until ready for use. A small section of each egg collar and sediment sample preserved in EtOH was weighed out. DNA was extracted with the NucleoSpin Soil DNA Isolation Kit (Takara; Kusatsu, Shiga, Japan) as instructed. DNA was extracted from all three of the egg collar sample thirds (per sample) and from each sediment sample. DNA concentration and quality were measured on a NanoDrop 2000 (ThermoFisher; Waltham, MA). DNA was stored at −20℃ until ready for use.

## 16S rRNA and adenylation domain (AD) gene amplification and sequencing

16S rRNA genes were amplified from environmental DNA using the universal V3/V4 16S primers (For: 5′– CCTACGGGNGGCWGCAG - 3′; Rev: 5′ – GACTACHVGGGTATCTAATCC – 3′). The ADs of NRPS enzymes were amplified using the following NRPS primers that were previously reported (F: 5′ – GCSTACSYSATSTACACSTCS GG – 3′; R: 5′ – SASGTCVCCSGTS CGGTA – 3′) (40). Amplified PCR products were sequenced at The Genomics Sequencing Center at Virginia Tech with Illumina MiSeq 600 cycle V3 chemistry to generate 50 million paired end reads (16S sequencing) or Illumina MiSeq V2 500 cycle Nano to generate 30 million paired end reads (NRPS sequencing).

## Identification of moon snail species

DNA extracted from egg collars collected in 2023 was amplified using mitochondrial 16S (M16S) (F: 5′ – CGCCTGTTACCAAAAACAT – 3'; R: 5′ - CCGGTCTGAACTCAGATCACGT – 3'), partial 18S (P18S) (F: 5′ – CGTGTTGATYCTGCCAGT – 3'; R: 5′ – TCTCAGGCTCCYTCTCCGG – 3′), and cytochrome oxidase subunit I (COXI) (F: 5′ – GCTTTTGTTATAATTTTYTT – 3'; R: 5′ – CGATCAGTTAAAARTATWGTAAT – 3') primers as described in (26, 26). Amplicons were cleaned using the ZymoResearch DNA Clean and Concentrator Kit following manufacturer's protocols. DNA concentration and quality were measured on a NanoDrop 2000 and sequenced with the forward primers. Sequences for COXI, P18S, and M16S were downloaded from NCBI for *Neverita duplicata* (COXI: EU492409, P18S: EU492394, M16S: EU492397) and *N. delessertiana* (COXI: EU492413, P18S: EU492395, M16S: EU492403) and were aligned against 2023 egg collar sequences with ClustalW for moon snail species identification (Fig. S8).

## 16S metataxonimic analysis on moon snail egg collar and sediment samples

Paired end sequences were imported into QIIME2 (v. 2020.6) as PairedEndFastqManifesPhred33 formatted samples. Sequences were denoised, and demultiplexed and chimera sequences were removed with DADA2 pipeline. Quality filters "--p-trim-left-f", "--p-trim-left-r", "--p-trunc-len-f", "--p-trunc-len-r", "--p-max-ee-f," and "--p-max-ee-r" were set to 10, 10, 200, 200, 2.0, and 2.0 respectively. Chimeras were removed with the consensus method ("--p-chimera-method consensus"), and threads were set to eight ("--p-n-threads 8"). SILVA v138.1, clustered at 99% identity, was used as the reference database for picking OTUs. ASV tables were generated in R (v. 4.1.0-foss-2021a) using the DADA2 (v. 1.22.0) pipeline and taxonomy was assigned with SILVA v138.1.

## NRPS analysis and annotation

Paired end NRPS sequences were imported into QIIME2 and demultiplexed as above in the integrated DADA2 pipeline. Quality filters "--p-trim-left-f", "--p-trim-left-r", "--p-trunc-len-f," and "--p-trunc-len-r" were set to 10, 10, 200, and 120 respectively. The demultiplexed sequences were aligned as described in (40) with the following exceptions—MIBiG data repository version was 3.1 (October, 2022), an e-value of $<10e^{-38}$ was used as a cut off value (40).

## Abundance plotting

The total abundance of all sequences across all samples was calculated, and the 20 most abundant were identified. The data frame was reshaped with reshape2 (v. 1.4.4), a facet sequence was generated, and the relative abundance for samples collected from each site was plotted using ggplot2 (v. 3.4.3). The average abundance data for the 50 most abundant sequences were log transformed and plotted with pheatmap (v. 1.0.12) using Euclidian distances. Boxplots were generated by plotting abundance counts for each sample with ggplot2 and gridExtra (v. 2.3). The viridis package (v. 0.6.4) was used to pick a color scheme; dplyr (v. 1.1.4) and tidyr (v. 1.3.0) were used for data manipulation (filtering, arranging, etc.) and data cleaning, respectively.

## Statistical analysis

Alpha diversity indices (Hill distances, rarefaction species richness) were calculated using the hillR (v. 0.5–2) and vegan (v. 2.6–4) R packages, respectively. Plotrix (v. 3.8.2) was used to calculate the standard error for species richness and plotted with ggplot2 and gridExtra. To visualize the dissimilarity between samples, the metaMDS function from vegan was used to calculate NMDS values using Bray–Curtis distances, then plotted with ggplot2. The adonis function in vegan was used to determine permutational multivariate analysis of variance (PERMANOVA) values between groups. ASV enrichment between egg collar and sediment samples was determined using DESeq2 (v. 1.38.3) and plotted with EnhancedVolcano (v. 1.16.0). Fold-change, abundance change, and abundance values were analyzed with the Shapiro–Wilk test for normality from the R base stats package. Statistical significance was calculated with the Kruskal–Wallis test, Welch two-sample *t*-test, and the Wilcoxon rank-sum test for fold-change, abundance change and abundance values, respectively.

## MIBiG Alignment and NCBI Blast

AD sequences obtained from Illumina sequencing were aligned with the MIBiG database (downloaded from https://mibig.secondarymetabolites.org/, version 3.1) using blastx with a 50% minimum sequence identity cutoff. AD sequences were then aligned against the NCBI non-redundant protein database with the blastx alignment function and filtered to remove matches with e-values $>10e^{-38}$. Genomes were obtained from gene accession numbers, analyzed with AntiSMASH, and corresponding BGCs were identified through ClustalW alignment between BGC ADs and AD sequences.

## ACKNOWLEDGMENTS

This work was funded by NIH [R35 GM146740 (E.M.)] and Virginia Tech Startup funds (E.M.). We thank the Virginia Tech Genomic Sequencing Center for sequencing services. We also thank S. and S. Mevers for collection assistance.

## AUTHOR AFFILIATIONS

[1]Department of Chemistry, Virginia Tech, Blacksburg, Virginia, USA
[2]Department of Biological Sciences, Virginia Tech, Blacksburg, Virginia, USA

## AUTHOR ORCIDs

Karla Piedl  http://orcid.org/0009-0002-4110-6269
Frank O. Aylward  http://orcid.org/0000-0002-1279-4050
Emily Mevers  http://orcid.org/0000-0001-7986-5610

## FUNDING

| Funder | Grant(s) | Author(s) |
|---|---|---|
| HHS | National Institutes of Health (NIH) | R35 GM146740 | Emily Mevers |

## AUTHOR CONTRIBUTIONS

Karla Piedl, Conceptualization, Data curation, Formal analysis, Methodology, Writing – original draft, Writing – review and editing | Frank O. Aylward, Conceptualization, Formal analysis, Methodology, Writing – review and editing | Emily Mevers, Conceptualization, Data curation, Funding acquisition, Methodology, Project administration, Writing – original draft, Writing – review and editing

## DATA AVAILABILITY

All 16S rRNA and NRPS raw sequences from the 2023 collection trip, 2024 collection trip and the NRPS study can be found in the SRA database under BioProject accession numbers PRJNA1133164, PRJNA1133692, and PRJNA1133717, respectively. COXI, M16S, and P18S sequences amplified from moon snail DNA can be found in the GenBank database under accession numbers PQ256765 (COXI), PQ256766 (COXI), PQ256767 (COXI), PQ256768 (COXI), PQ255925 (M16S), PQ255926 (M16S), PQ255927 (M16S), PQ255928 (M16S), PQ255929 (P18S), PQ255930 (P18S), PQ255931 (P18S), and PQ255932 (P18S).

## ADDITIONAL FILES

The following material is available online.

### Supplemental Material

**Supplemental figures and tables (Spectrum01804-24-s0001.docx).** Tables S1 to S11; Fig. S1 to S8.

### Open Peer Review

**PEER REVIEW HISTORY (review-history.pdf).** An accounting of the reviewer comments and feedback.

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
