## [Reviewer comments · Microbiology Spectrum]

Microbiology Spectrum

The Microbiota of Moon Snail Egg Collars is Shaped by Host-Specific Factors

Karla Piedl, Frank Aylward, and Emily Mevers

Corresponding Author(s): Emily Mevers, Virginia Polytechnic Institute and State University

Review Timeline:

Submission Date:	July 19, 2024
Editorial Decision:	August 21, 2024
Revision Received:	September 3, 2024
Accepted:	September 4, 2024

Editor: Jan Claesen

Reviewer(s): The reviewers have opted to remain anonymous.

Transaction Report:

DOI: <https://doi.org/10.1128/spectrum.01804-24>

Re: Spectrum01804-24 (The Microbiota of Moon Snail Egg Collars is Shaped by Host-Specific Factors)

Dear Dr. Emily Mevers:

Thank you for the privilege of reviewing your work. Below you will find my comments, instructions from the Spectrum editorial office, and the reviewer comments.

Thanks for submitting your research to Spectrum. Your manuscript has been evaluated by an expert Reviewer and they are excited about your work (as am I). There are some comments and suggestions listed below which would help improve the manuscript and I would be happy considering a revised version that addresses these in a point-by-point manner.

Revision Guidelines

Sincerely,
Jan Claesen
Editor
Microbiology Spectrum

Reviewer #1 (Comments for the Author):

This paper characterizes the microbiome associated with moon snail egg collars along with adenylation domains associated with the production of biosynthetic compounds. This work establishes an important baseline for developing the moon snail microbiome as a model for studying egg-associated microbiota and defensive symbioses. The manuscript is clearly written and

will appeal to those studying host-microbe interactions and natural products chemistry. I do have some comments and suggested edits before publication:

Intro and/or discussion: I suggest the authors include more background information about the biology of moon snails, especially their development. I think it's important to try and correlate the different egg collar stages that the authors describe (i.e., firm, less firm, crumbly, and fragile) with the timing of snail embryogenesis if possible.

Also, if the egg collar bacteria are transmitted from the parent snails (as stated in the conclusions line 392-393), what organs or structures do the hosts have to facilitate deposition; do these harbor bacteria? An alternate hypothesis may also be that that egg mucus environment selects and enriches for the bacteria from the environment, and it seems like this is what is being suggested in lines 349-352. Both hypotheses are plausible but I would make sure each is clearly stated and discussed.

other comments:

Abstract and Methods: Please include the scientific name for the moon snail that was sampled in this study.

Line 43 "are" to "is"

Line 90: antibiotic treatment of eggs leading to fungal infection in bobtail squid should be reference Kerwin et al., 2019 PMID: 31662458.

Line 98: in addition to reference 20: Kerwin et al., 2019 PMID 3162458.

Line 124: "mucky". Do the authors mean muddy?

Lines 159-162: (please see my comments above) Is there literature on the embryogenesis of moon snails? How long do they take to develop? This might provide a range of specific developmental stages of the egg collars that correspond to firm and fragile descriptions.

Line 214: OUT should be OTU?

Line 309-310: What genomes were downloaded for analyses? I didn't see this information in the methods or supplemental info. Is this indicated in Table 2?

Line 353: "other cephalopods" This may just be a wording issue, but to some readers it might imply that moon snails are cephalopods. I would edit to "other molluscs" or rephrase this sentence, to something like "compared to other molluscan eggs (e.g., cephalopods) or similar.

Discussion: Do the authors know how the moon snail egg collar microbiome compares to other marine gastropod microbiome analyses?

Discussion: In the introduction, it's stated that the authors have more than 700 bacterial strains in culture from moon snail egg collars. Is there genomic information from these that could be used to search for the AD domain genes and other biosynthetic gene clusters? A more focused or targeted approach may reveal AD-containing BGCs and pathways that are important in moon snail eggs as opposed to searching publicly available genomes that are from other sources. If genomes from these isolates are not available, it might be a good point of discussion for future directions.

Lines 380-382: The presence of the AD genes doesn't necessarily mean that there is expression or production of or specific compounds produced in the egg collars. A future direction might be to look at gene expression or protein production of specific factors involved with BGC production.

Lines 392-393. Please see my comment above about an alternate hypothesis for bacterial selection/enrichment in the egg collars.

Reviewer #1 (Comments for the Author):

This paper characterizes the microbiome associated with moon snail egg collars along with adenylation domains associated with the production of biosynthetic compounds. This work establishes an important baseline for developing the moon snail microbiome as a model for studying egg-associated microbiota and defensive symbioses. The manuscript is clearly written and will appeal to those studying host-microbe interactions and natural products chemistry. I do have some comments and suggested edits before publication:

Intro and/or discussion: I suggest the authors include more background information about the biology of moon snails, especially their development. I think it's important to try and correlate the different egg collar stages that the authors describe (i.e., firm, less firm, crumbly, and fragile) with the timing of snail embryogenesis if possible.

Response: *We thank the reviewer for this insightful comment. More background focusing on the development and embryogenesis of moon snails has been added to the introduction and discussion sections.*

Also, if the egg collar bacteria are transmitted from the parent snails (as stated in the conclusions line 392-393), what organs or structures do the hosts have to facilitate deposition; do these harbor bacteria? An alternate hypothesis may also be that that egg mucus environment selects and enriches for the bacteria from the environment, and it seems like this is what is being suggested in lines 349-352. Both hypotheses are plausible but I would make sure each is clearly stated and discussed.

Response: *We apologize as it was not our intent to assert that the bacteria are transmitted from the parent snail but rather that this is just one of several possibilities. We have rephrased to indicate other possibilities, including the possibility that the nutrient profile of the mucus selects for specific bacteria*

other comments:

Abstract and Methods: Please include the scientific name for the moon snail that was sampled in this study.

Response: *We actually received sequencing data to identify the moon snail down to the species level - *Neverita delessertiana*. We have updated the manuscript accordingly.*

Line 43 "are" to "is"

Response: *Corrected as suggested.*

Line 90: antibiotic treatment of eggs leading to fungal infection in bobtail squid should be reference Kerwin et al., 2019 PMID: 31662458.

Response: *We have added this reference and thank the reviewer for pointing out our mistake.*

Line 98: in addition to reference 20: Kerwin et al., 2019 PMID 3162458

Response: *Corrected as suggested.*

Line 124: "mucky". Do the authors mean muddy?

Response: *Corrected as suggested.*

Lines 159-162: (please see my comments above) Is there literature on the embryogenesis of moon snails? How long do they take to develop? This might provide a range of specific developmental stages of the egg collars that correspond to firm and fragile descriptions.

Response: *Surprisingly, there is very little information about embryogenesis of moon snail eggs; however reanalysis of the current literature did reveal a couple of papers that we previously overlooked. We used this new information to add relevant details about embryogenesis and moon snail life cycle in general to the introduction.*

Line 214: OUT should be OTU?

Response: *Corrected as suggested.*

Line 309-310: What genomes were downloaded for analyses? I didn't see this information in the methods or supplemental info. Is this indicated in Table 2?

Response: *We added text to clarify that the accession numbers for downloaded genomes can be found in Table 2. In addition, we included a footnote to clarify that all genomes listed in that column were downloaded and analyzed.*

Line 353: "other cephalopods" This may just be a wording issue, but to some readers it might imply that moon snails are cephalopods. I would edit to "other molluscs" or rephrase this sentence, to something like "compared to other molluscan eggs (e.g., cephalopods) or similar.

Response: *The text was updated to clarify that moon snails are not cephalopods, but are from the same phylum*

Discussion: Do the authors know how the moon snail egg collar microbiome compares to other marine gastropod microbiome analyses?

Response: *Unfortunately, not much is known about other marine gastropod microbiomes. We found a paper by Daniels, B.N., et al., *Microbiol. Spect.* 2024 (DOI: [10.1128/spectrum.03514-23](https://doi.org/10.1128/spectrum.03514-23)), that focused on whelk egg microbiomes, but have been unable to find any other reports that used molecular taxonomy techniques. We have updated the discussion section to include this information.*

There are reports of microbiomes of other snails using culture-based approaches but we do not believe it is appropriate to draw comparisons between these studies and our findings as culture-based approaches tend to poorly represent bacterial microbiomes.

Discussion: In the introduction, it's stated that the authors have more than 700 bacterial strains in culture from moon snail egg collars. Is there genomic information from these that could be used to search for the AD domain genes and other biosynthetic gene clusters? A more focused or targeted approach may reveal AD-containing BGCs and pathways that are important in moon snail eggs as opposed to searching publicly available genomes that are from other sources . If

genomes from these isolates are not available, it might be a good point of discussion for future directions.

Response: *Unfortunately, we have only sequenced the genomes of a small fraction (~36) of our strain library. Of these sequenced genomes, only 3 belong to the Flavobacteriia. We did perform an alignment of the AD sequences against our sequenced genomes but did not find any matches. We have updated the discussion to include a sentence about searching for AD domains in our sequenced genomes.*

Lines 380-382: The presence of the AD genes doesn't necessarily mean that there is expression or production of or specific compounds produced in the egg collars. A future direction might be to look at gene expression or protein production of specific factors involved with BGC production.

Response: *We have clarified the text to reflect that presence of the AD sequences does not mean production of specific compounds, but rather that there is the potential to produce NRPS-like compounds on the egg collars.*

Lines 392-393. Please see my comment above about an alternate hypothesis for bacterial selection/enrichment in the egg collars.

Response: *Similar to our response above, we have softened the language around how the egg collars acquire their microbiome components.*

Re: Spectrum01804-24R1 (The Microbiota of Moon Snail Egg Collars is Shaped by Host-Specific Factors)

Dear Dr. Emily Mevers:

Thank you for carefully addressing the Reviewer comments. I'm excited to congratulate you on the acceptance of your paper for publication in Spectrum!

Your manuscript has been accepted, and I am forwarding it to the ASM production staff for publication. Your paper will first be checked to make sure all elements meet the technical requirements. ASM staff will contact you if anything needs to be revised before copyediting and production can begin. Otherwise, you will be notified when your proofs are ready to be viewed.

Sincerely,
Jan Claesen
Editor
Microbiology Spectrum